# Novel High-Throughput Microwell Spectrophotometric Assay for One-Step Determination of Lorlatinib, a Novel Potent Drug for the Treatment of Anaplastic Lymphoma Kinase (ALK)-Positive Non-Small Cell Lung Cancer

**DOI:** 10.3390/medicina59040756

**Published:** 2023-04-13

**Authors:** Abdullah M. Al-Hossaini, Ibrahim A. Darwish, Hany W. Darwish

**Affiliations:** Department of Pharmaceutical Chemistry, College of Pharmacy, King Saud University, P.O. Box 2457, Riyadh 11451, Saudi Arabia

**Keywords:** 2,3-dichloro-3,5-dicyano-1,4-benzoquinone, lorlatinib, non-small cell lung cancer, charge transfer complex, spectrophotometry, high throughput

## Abstract

*Background and Objectives:* Lorlatinib (LOR) belongs to the third-generation anaplastic lymphoma kinase (ALK) tyrosine kinase inhibitors. People who are diagnosed with ALK-positive metastatic and advanced non-small cell lung cancer (NSCLC) are eligible to get it as a first-line treatment option after it was given the approval by “the Food and Drug Administration (FDA)”. However, no study has described constructing high-throughput analytical methodology for LOR quantitation in dosage form. For the first time, this work details the construction of a high-throughput, innovative microwell spectrophotometric assay (MW-SPA) for single-step assessment of LOR in its tablet form, for use in pharmaceutical quality control. *Materials and Methods:* Assay depended on charge transfer complex (CTC) formation between LOR, as electron donor, with 2,3-dichloro-3,5-dicyano-1,4-benzoquinone (DDQ), as π-electron acceptor. Reaction conditions were adjusted, the CTC was characterized by ultraviolet (UV)-visible spectrophotometry and computational molecular modeling, and its electronic constants were determined. Site of interaction on LOR molecule was allocated and reaction mechanism was suggested. Under refined optimum reaction conditions, the procedures of MW-SPA were performed in 96-well assay plates, and the responses were recorded by an absorbance plate reader. Validation of the current methodology was performed in accordance with guidelines of “the International Council on Harmonization (ICH)”, and all validation parameters were acceptable. *Results:* Limits of detection and quantitation of MW-SPA were 1.8 and 5.5 µg/well, respectively. The assay was applied with great success for determining LOR in its tablets. *Conclusions:* This The assay is straightforward, economic and has high-throughput characteristics. Consequently, the assay is recommended as a valuable analytical approach in quality control laboratories for LOR’s tablets’ analysis.

## 1. Introduction

Because it is primary main cause of death in both sexes all over world, cancer is a serious global health problem that has to be addressed. In the year 2020, it was anticipated that 18.1 million people would be diagnosed with cancer worldwide. When considering all forms of cancer, lung cancer is the one that occurs in the second most people around the world. It is most frequent kind of cancer to affect men and second most common form of cancer to affect women. In the year 2020, there were more than 2.2 million newly diagnosed cases of lung cancer which makes it the largest cause of cancer-related mortality [1] and is responsible for the most deaths. It is important to note that lung cancer is subdivided into small-cell and non-small-cell lung cancer (NSCLC). NSCLC is responsible for roughly 80–85 percent of the total cases of lung cancer. Main treatment options for NSCLC include surgery, radiotherapy, and chemotherapy. For localized stage I and stage II of NSCLC, surgery and radiotherapy are usually useful; however, they are not authorized for advanced stages of NSCLC [2,3]. For ~80% of all patients with NSCLC, chemotherapy is usually recommended as a superior first-line treatment regime because It has been shown to increase both longevity and quality of life [4,5,6].

First-generation tyrosine kinase inhibitors such as gefinitinib, erlotinib, and other have been widely prescribed [7,8]; however, a large proportion of patients have a chromosomal rearrangement that results in a fusion gene for anaplastic lymphoma kinase (ALK) and echinoderm microtubule-associated protein like 4 (EML4). The created gene causes constitutive activity of the kinase protein, and ultimately leads to increased cell carcinogenesis and induces malignant phenotype [9,10,11]. First generation tyrosine kinase inhibitors are unable to hinder fusion protein kinase activity. Therefore, drug discovery researchers were directed to discover novel drugs for treating patients with NSCLC including those harbor EML4/ALK proteins. The efforts of the researchers ultimately lead to the emerging of most potent third generation ALK inhibitor drugs, which gave a special interest and clinical benefits in treatment of ALK-positive patients NSCLC who are not responding to the earlier generations of ALK inhibitors.

Lorlatinib (LOR) is the first member of the third generation of ALK inhibitors [12,13]. Because of the clinical success of LOR, US-FDA on 3 March 2021, has been approved LOR for patients with metastatic NSCLC whose tumors are ALK-positive, distinguished by an FDA-approved test [14]. LOR was also approved by the European Medicines Agency (EMA) for treating patients with advanced and ALK-positive NSCLC [15]. LOR is sold in the markets under the trade name of Lorbena^®^ tablets “Pfizer Inc., New York, NY, USA”. Recommended dose of LOR is 100 mg orally once daily [16].

The therapeutic benefits of LOR, in terms of its potency and safety, is principally depending on pharmaceutical formulation quality (Lorbrena^®^ tablets). Also, the therapeutic success of LOR is expected to be encouraging to the pharmaceutical companies for the development of new pharmaceutical preparations for LOR once the patent of Lorbrena^®^ tablets for Pfizer expires. Therefore, LOR determination in its pharmaceutical tablets would be necessary. To achieve this goal, a proper analytical method with high throughput is vital. Literature review revealed that only one analytical technology exists for determination of LOR; this technology is liquid chromatography [17,18,19,20,21,22]. Most of the existing chromatographic methods [18,19,20,21,22] relied on the expensive and instrumental intensive tandem mass spectrometric detectors and devoted to the analysis of LOR in biological samples. Only one chromatographic method is published for determining LOR in its bulk and pharmaceutical formulation [17]. The standards approach for quality control of pharmaceuticals require simple and high-throughput procedures, rather than the chromatographic procedures which usually have limited throughput. Because it allows the analysis of dozens, hundreds, or even thousands of samples every day in a given laboratory or on a specific instrument, high-throughput analysis is becoming increasingly significant in the pharmaceutical sector. This approach is important in the speedy identification of pharmaceutical entities, testing the uniformity of pharmaceutical formulations [23,24,25].

This manuscript designates, for the first time, development and validation of new microwell spectrophotometric assay (MW-SPA) with high-throughput for determination of LOR in pharmaceutical quality control laboratories. MW-SPA, described herein, is based on formation of colored charge-transfer complex (CTC) between LOR (electron donor), 2,3-dichloro-5,6-dicyano-1,4-benzoquinone (DDQ) (π-electron acceptor). Procedures of MW-SPA are performed in 96-well assay plates and absorbances are recorded by absorbance plate reader.

## 2. Experimental

### 2.1. Apparatus

A double-beam ultraviolet-visible spectrophotometer (V-530: JASCO Co., Ltd., Kyoto, Japan) was utilized for recording absorption spectra. Absorbance microplate reader (ELx808: Bio-Tek Instruments Inc., Winooski, VT, USA) empowered by KC Junior software, provided with the instrument. The reader adapts a top-reading mode, and it is also equipped with a bult-in heating system which enables the control of the temperature and maintain the temperature of the assay plates at the desired temperature. It can also apply a shaking action for 1 min for mixing the reagent with the analyte and settle a brief time (300 ms) after shaking before measuring the absorbances.

### 2.2. Chemicals and Materials

Lorlatinib (LOR) was purchased from “Selleck Chemicals (Houston, TX, USA)”. Purity was claimed to be 99.73% and solution stability was assured for a minimum of seven days on refrigeration. Lorbrena^®^ tablets “Pfizer Inc., New York, NY, USA” were kindly donated by SFDA (Riyadh, Saudi Arabia) and its potency was claimed to be 100 mg of LOR per tablet. DDQ was purchased from “Sigma-Aldrich Chemicals Co. (St. Louis, MO, USA)”; its solution (0.4%, *w*/*v*, in methanol) was freshly prepared. Corning^®^ 96–well transparent polystyrene assay plates with flat bottoms were purchased from Merck & Co., Inc. (Rahway, NJ, USA). Adjustable single and 8-channel pipettes (Finnpipette^TM^) were obtained from Thermo Fisher Scientific Inc. (Waltham, MA, USA). Reagent reservoirs (BRAND^®^ PP) with cover lids for dispensing the solutions by the 8-channel pipettes were purchased from “Merck KGaA (Darmstadt, Germany)”. Reagents and solvents were of spectroscopic grade “Fisher Scientific, California, CA, USA”.

### 2.3. Preparation of Standard LOR Solutions

LOR stock solution was prepared by dissolving 20 mg (8.13 × 10^−2^ mole) of LOR in 10 mL methanol. This stock solution (4.9 × 10^−3^ M) was diluted with methanol to acquire working solutions of LOR with concentrations suitable for the corresponding study.

### 2.4. Preparation of Tablets Sample Solution

Ten lorbrena^®^ tablets were ground into a powdered form. In a 50-mL calibrated flask, a precise amount of the fine powder equal to 100 mg of LOR was transferred. Approximately 15 mL of methanol was added, and the mixture was vigorously shaken to dissolve LOR completely. The contents of the flask were filtered out, and the initial fraction of the filtrate was omitted. Filtrate was then diluted with methanol to yield 50–2000 μg mL^−1^ LOR solutions. These solutions were analyzed for their nominated LOR concentrations by MW-SPA.

### 2.5. Association Constant and Molar Ratio Calculation

The association constant of LOR-DDQ CTC and its molar ration were determined according to the methods described by Benesi-Hildebrand [26] and Job [27], respectively. For calculation of the association constant, a series of LOR solutions of varying concentrations (7.66 × 10^−5^–9.23 × 10^−4^ M) were equilibrated for ~5 min at 25 ± 2 °C with fixed concentration of DDQ (5.51 × 10^−3^ M). Absorbances of the mixed equilibrated solutions were exploited for Benesi-Hildebrand plot generation. The results of regression analysis of the plot were used for computing the association constant.

For determination of LOR-DDQ ratio by “Job’s method” [27], equimolar solutions (4.9 × 10^−3^ M) of LOR and DDQ reagent were used to make up different complementary reaction proportions (0:00, 25:175, 50:150, 100:100, 125:75, 150:50, 175:25, and 200:0). The measured absorbances were graphed versus [LOR]/[LOR + DDQ] and molar ratio of LOR/DDQ reaction was calculated using the generated plot.

### 2.6. Procedure of MW-SPA

Wells of 96-microwell assay plates were dispensed with aliquots (100 µL) of standard or tablet sample solution containing different concentrations of LOR (5–200 g). After addition of 100 µL of a DDQ solution (0.4%, *w*/*v*), reaction was left to run for 5 min at 25 ± 2 °C. Absorbance microplate reader was used to measure responses at 460 nm.

## 3. Results and Discussion

### 3.1. Methodology and Strategy for Assay Development

Spectrophotometric assays are widely used in pharmaceutical industries [28,29,30]. These assays gained their importance and wide applications because they can be readily automated with spectrophotometric analyzers, that enable the analysis of hundreds of samples for the quality control assessment of pharmaceutical formulations, such as testing uniformity of contents and dissolution testing of tablets. The CT reactions have been widely employed as a basis for developing many spectrophotometric assays for the purpose of quality control of pharmaceuticals [31,32,33,34]. These assays are simple and rapid. Obviously, the chemical structure of LOR contains multiple electron-donating sites (e.g., amino group nitrogen atoms and ether-oxygen atom), which may contribute to the formation of CTC with electron acceptors. As revealed from an intensive literature review, CT reaction of LOR was not published yet. Therefore, our study is considered the first methodology in this area. DDQ is a versatile reagent which has been successfully utilized in varied chemical reactions and applications [35,36]. DDQ is commercially available, user-friendly, shelf-stable, solvent-soluble, and frequently produces non-toxic byproducts during its reactions. Alzoman et al. [37] established that DDQ is one of the most reactive acceptors in a prior work incorporating CT reactions with numerous polyhalo-/polycyanoquinone electron acceptors, and as a result, it has been widely used in the creation of several CT-based spectrophotometric tests for pharmaceuticals [37,38,39,40]. Due of these factors, DDQ was chosen as the electron acceptor for the MW-SPA presented here.

The conventional practice of the spectrophotometric assays, including CT reactions, usually employ volumetric flasks/cuvettes in the analysis, thus these assays have limited throughputs that do not meet the needs of quality control laboratories for the processing of pharmaceutical formulations [23,24,25]. In addition, the conventional CT-based assays consume large volumes of costly and more prominently it may cause analysts’ toxicity [41,42,43,44]. As a result, developing new methodologies of higher throughput and utilizing small volumes of organic solvents was considered in constructing CT-based assays. Recently, our laboratory successfully employed absorbance microplate reader in adopting MW-SPA for quantitation of active constituents in their dosage forms [45]. These assays provided high throughput analysis and consumed low organic solvents volumes. For these reasons, the present study has undertaken the development of this methodology for LOR.

### 3.2. Absorption Spectra and Band Gap Energy

An LOR solution UV-visible absorption spectra (4.92 × 10^−4^ M, in methanol) was measured (Figure 1) and it displayed existence of two absorption maxima (λ_max_) at 298 and 390 nm. When different concentrations of LOR solutions (0.77 × 10^−4^–9.23 × 10^−4^ M) were mixed with a fixed concentration of DDQ solution (5.51 × 10^−3^ M), the reaction between LOR and DDQ was permitted to occur at 25 ± 2 °C, and reaction mixture absorption spectra were measured vs. a DDQ reagent blank solution (Figure 1). Red color product with absorbance maximum of 460 nm was obtained and its intensity grew as the LOR concentration in reaction solution increased. This characteristic was clearly observed throughout the process of producing a new product from a reaction. The shape and pattern of resulting absorption band was comparable to DDQ radical anion as detailed in literature [37,38,39,40]. Hence, reaction was assumed to be a CT reaction involving LOR (electron donor, D) and DDQ (π-electron acceptor, A), and reaction continued in methanol (as an example of polar solvent) to create a CT complex (D-A), which was then dissociated by methanol giving radical anion of DDQ (A^•−^), as shown in Figure 1A.

Band gap energy (Eg), defined as least amount of energy essential to excite an electron and cause its transition from low-energy valence band to the high-energy one and contribute to the formation of the formation of the CTC band, was determined according to procedures designated by Karipcin et al. [46]. Eg value was 2.7036 eV. This small value indicated easiness of electron transfer from LOR to DDQ.

### 3.3. Reaction Conditions’ Optimization

For determine utmost suitable solvent for optimal reaction, reaction between LOR and DDQ was performed in many solvents possessing different dielectric constants [47] and polarity indexes [48]. Results indicated that slight shifts in λ_max_ accompanied with changes in molar absorptivity (ε) occurred. Reaction in solvents possessing both high dielectric constants and polarity (methanol and acetonitrile) gave better ε values than those in solvents with low dielectric constants and polarity (diethyl ether and dichloroethane) did. This effect was explained by the fact that the process of transferring electron from electron donor molecule (LOR) to electron acceptor (DDQ) is favored in polar solvents and diminished in non-polar solvents. In all subsequent steps, methanol was utilized as the proper solvent. At 25 ± 2 °C the reaction in methanol was immediate. The effect of DDQ concentration was studied by performing reaction using varying concentrations of DDQ (0.1–1%, *w*/*v*). Results revealed that color intensity of CTC increases as the DDQ concentration increases and the maximum color intensity was achieved when the DDQ concentration ranged from 0.25 to 0.5% (*w*/*v*), beyond which the color intensity decreased (Figure 2). Accordingly, DDQ at a concentration of 0.4% (*w*/*v*) was utilized in succeeding practical work. A summary of optimum reaction conditions was given in Table 1.

### 3.4. Electronic Constants and Properties

The relevant electronic constants and properties of the LOR-DDQ CTC were determined according to the methods described in previous reports [26,49,50,51,52,53]. These constants and properties were the association constant [26], derived from Benesi-Hildbrand plot (Figure 3), standard free energy change [49], ionization potential [50], resonance energy [51], oscillator strength [52], and transition dipole moment [53]. Determination methods and obtained values were summarized in Table 2. Obviously, the big value of the association constant and low value of standard free energy change suggest that interaction between LOR and DDQ done simply and formed LOR-DDQ CTC was adequately stable. In addition, the high value of molar absorptivity of CTC enables the development of a sensitive assay for LOR.

### 3.5. Molar Ratio and Computational Charge Calculation

Using Job’s continuous variation approach [27], molar ratio of LOR to DDQ was determined to be 1:1 (Figure 4), showing that only one site of LOR participated in creating CTC with DDQ. To identify this site among the several accessible electron-donating sites on LOR molecule (Figure 5), energy minimization was conducted on LOR molecule and electron density on each atom was determined. Energy minimization as well as charge calculation were carried out utilizing “CS Chem3D Ultra, version 16.0 (CambridgeSoft Corporation, Cambridge, MA, USA)” in conjunction with “molecular orbital computations software (MOPAC) and molecular dynamics computations software (MM2 and MMFF94)”. Outcomes are shown in Table 3. Number 25 nitrogen atoms were assigned to have highest electron density (N25: aniline nitrogen). Taking into mind the molar ratio (1:1), it was hypothesized that the CT reaction between LOR and DDQ would proceed as seen in Figure 6.

### 3.6. Development of MW-SPA

The optimum experimental conditions for carrying out reaction between LOR and DDQ in the 96-microwell assay plate were summarized in Table 1. The absorbances were measured by the absorbance microplate reader at 450 nm (nearest filter to λ_max_ of CTC of LOR with DDQ, 460 nm).

### 3.7. Validation

#### 3.7.1. Linearity and Sensitivity

Under MW-SPA-optimized settings, calibration curve (Figure 7) was constructed, then linear regression analysis of the dataset was performed utilizing least-squares approach. Within the range of 5 to 200 μg/well (100 μL), curve was linear, with 0.9996 correlation coefficient. Intercepts, slopes, and correlation coefficients (the three main components of linear fit) are displayed in Table 4. According to ICH criteria [54], LOD and LOQ were determined. LOD and LOQ values were 1.8 and 5.5 μg/well, respectively. These limits are adequate for quantitation of LOR in only tablet’s dosage form but not in biological fluids. Table 4 provides a brief overview of proposed methodology’s calibration and validation parameters.

#### 3.7.2. Precision and Accuracy

Samples of LOR solutions with varied concentrations were utilized to ensure precisions of the proposed MW-SPA (Table 5). For intra– and inter–assay precision, RSD ranged from 0.15 to 1.76 and 1.12 to 1.86%, respectively. It was clear from small RSD values that the assay was highly precise. At the same levels of LOR concentrations as the precision experiments, assay’s accuracy was assessed by recovery studies. According to Table 5, recovery values ranged from 98.1 to 103.1%, demonstrating high accuracy of current assay.

#### 3.7.3. Specificity and Interference

Specificity of proposed MW-SPA for LOR quantitation in tablets without any potential interference from the inactive excipients which are co-formulated with LOR is documented because of two main reasons. The first one is the measurements of LOR in the visible region (460 nm) which is far away from the UV-absorbing excipients which might be extracted from the tablets during the preparation of tablets solution. The second reason is the use of methanol for preparation of tablet solution which dissolve only LOR leaving the excipients because they do not dissolve in methanol.

#### 3.7.4. Robustness and Ruggedness

MW-SPA robustness (impact of minor variation in assay variables on performance) was assessed. Concentration of DDQ, reaction time, and temperature were all altered by a factor of 10% from their optimum values (Table 1). Minor procedure variables didn’t disturb assay results revealing assay suitability for routine LOR analysis.

Two analysts performed the experiment on three different days to test ruggedness. Since the maximum RSD ≤ 2%, the results derived from differences in analysis from one analyst to another and from day to day were determined to be reproducible.

### 3.8. Analysis of Lorbrena^®^ Tablets

Successful validation results obtained depicted the appropriateness of current MW-SPA methodology for routine QC analysis of LOR-containing commercial pharmaceutical tablets. MW-SPA methodology was applied for assessing LOR in lorbrena^®^ tablets. Computed mean value of labelled amount was 101.3 ± 1.64% (Table 6).

## 4. Conclusions

UV-visible spectrophotometric analysis confirmed production of CTC between LOR and DDQ, as demonstrated by the emergence of a new absorption peak with maximal absorption peak at 460 nm. It was determined that molar ratio of CTC (LOR:DDQ) was 1:1. Molar absorptivity of CTC was dependent on both solvent’s polarity index and dielectric constant. Reaction mechanism was hypothesized. The reaction between LOR and DDQ served as the basis for the development of a unique MW-SPA for analyzing LOR in bulk and tablets. Assay offers a high throughput, permitting a large number of samples to be analyzed in a short amount of time. Additionally, when employed in quality control laboratories, the assay is eco-friendly because it employs a very small volume of organic solvent.

## Figures and Tables

**Figure 1 medicina-59-00756-f001:**
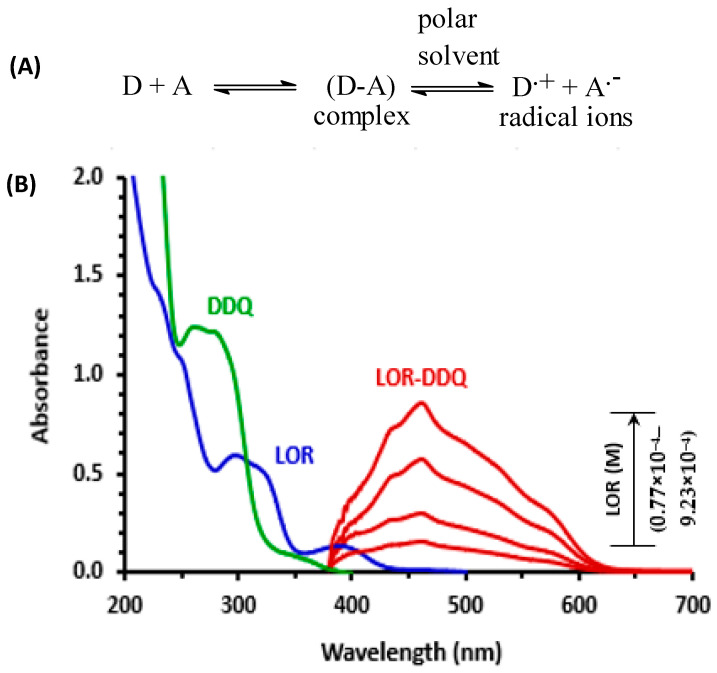
Panel (**A**): CT reaction between donor (D) and acceptor (A). Panel (**B**): Absorption spectra of lorlatinib, LOR (4.92 × 10^−4^ M), 2,3-dichloro-3,5-dicyano-1,4-benzoquinone, DDQ (5.51 × 10^−3^ M), and reaction mixtures (LOR-DDQ) containing varying concentrations of LOR (0.77 × 10^−4^ M–9.23 × 10^−4^ M) and a fixed concentration of DDQ (5.51 × 10^−3^ M); all solutions were in methanol.

**Figure 2 medicina-59-00756-f002:**
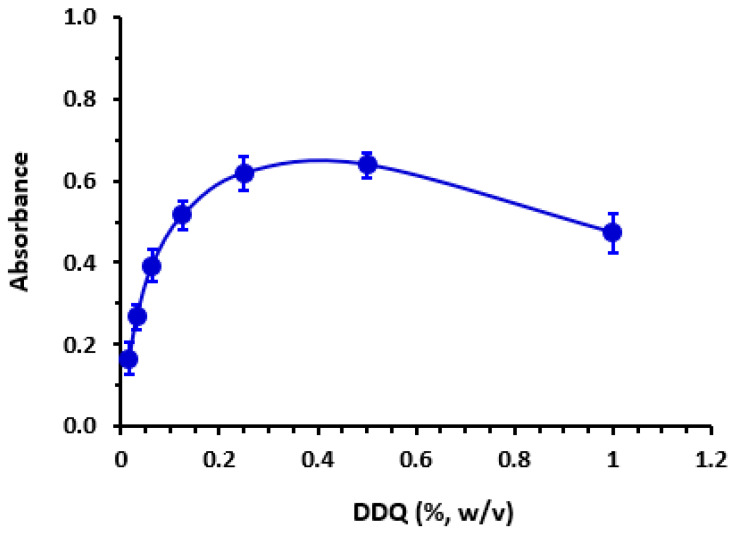
The effect of 2,3-dichloro-3,5-dicyano-1,4-benzoquinone (DDQ) concentration on charge transfer (CT) reaction with LOR (6.15 × 10^−4^ M, in methanol).

**Figure 3 medicina-59-00756-f003:**
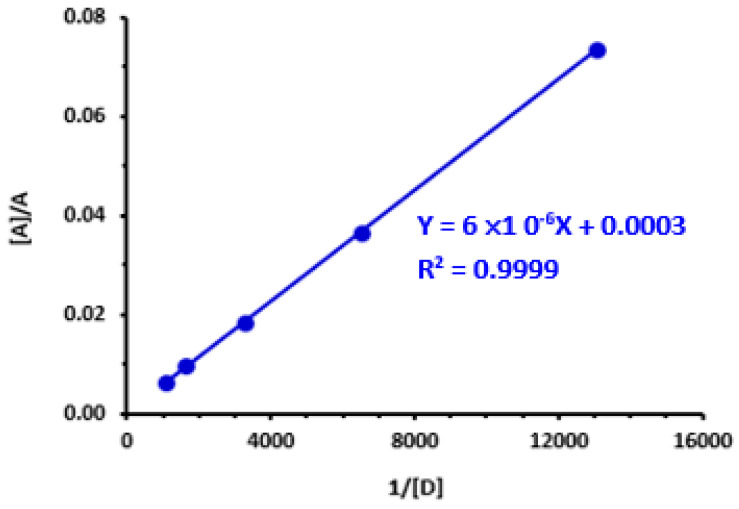
Benesi-Hildebrand plot for the formation of CTC of LOR with DDQ. Linear fitting equation and correlation coefficient (r^2^) are given on the plot. [A], A and [D] are molar concentrations of DDQ, absorbances of CTC, and molar concentration of LOR, respectively.

**Figure 4 medicina-59-00756-f004:**
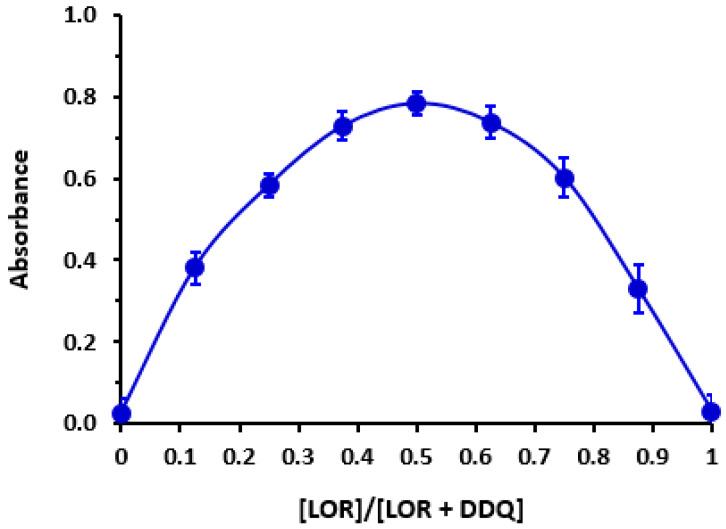
Job’s continuous variation plot for molar ratio determination of CT reaction of LOR with DDQ.

**Figure 5 medicina-59-00756-f005:**
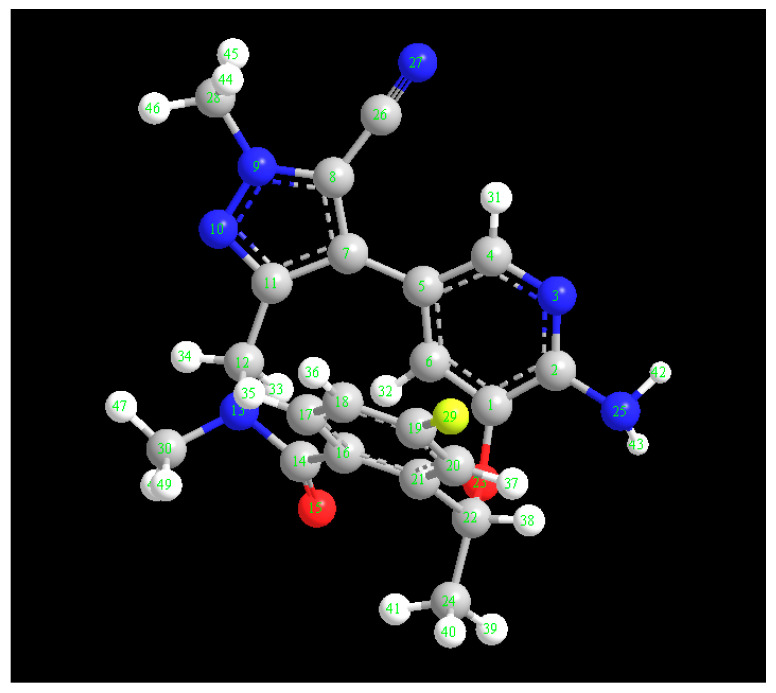
The energy minimized LOR molecule with atom numbers, and arrow points to atom wit the highest electron density (N25) which participated in CTC formation between LOR and DDQ.

**Figure 6 medicina-59-00756-f006:**
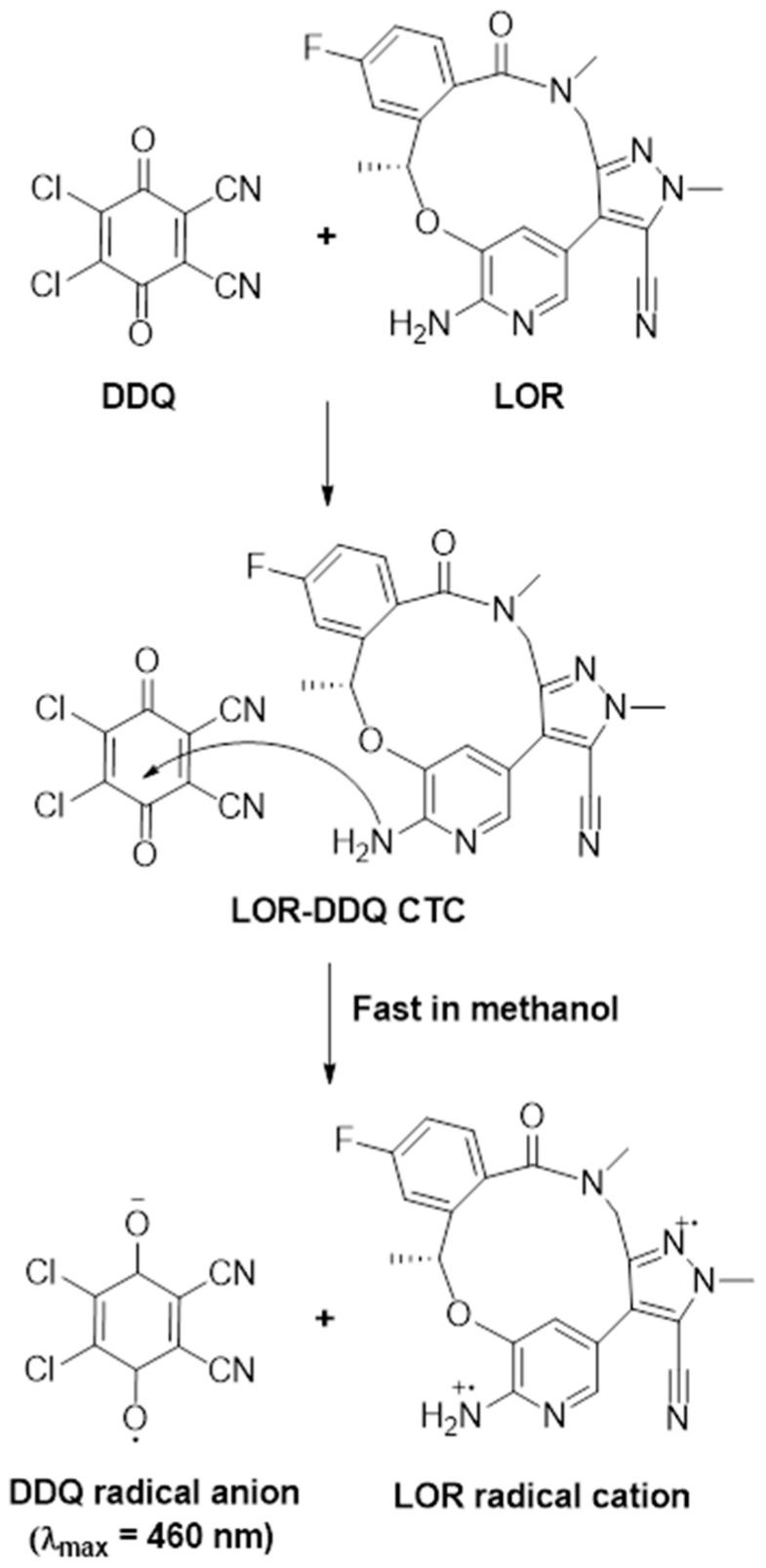
The scheme for the formation of the CTC between LOR and DDQ.

**Figure 7 medicina-59-00756-f007:**
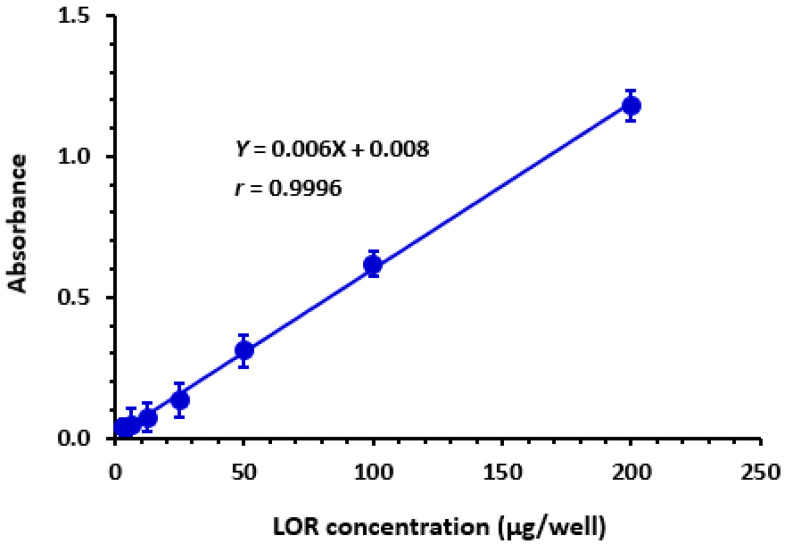
The calibration curve for the determination of LOR by MW-SPA via formation of CTC with DDQ. Linear regression equation and its correlation coefficient (*r*) is given on calibration line.

**Table 1 medicina-59-00756-t001:** Optimization of MW-SPA experimental conditions.

Condition	Investigated Range	Optimal Value
DDQ conc. (%, *w*/*v*)	0.1–1	0.4
Solvent	Different ^a^	Methanol
Reaction time (min)	0–30	Instantaneous ^b^
λ_max_ (nm)	350–750	460 ^c^

^a^ Solvents used were acetonitrile, methanol, ethanol, propanol, diethyl ether, dichloroethane, and chloroform. ^b^ Measurements on plate reader were performed within 5 min. ^c^ Measurements on plate reader were performed at 450 nm.

**Table 2 medicina-59-00756-t002:** Electronic constants/properties of CTC of LOR with DDQ.

Constant/Property	Method [Reference]	Equation/Procedure	Value
Molar absorptivity, ε (L mol^−1^ cm^−1^)	Benesi-Hildebrand [26]	Given in Section 2	0.34 × 10^3^
Association constant, *K* (L mol^−1^)	Benesi-Hildebrand [26]	The same equation used for calculating ε	0.52 × 10^2^
Ionization potential, I_p_ (eV)	Aloisi and Piganatro [50]	I_p_ (eV) = 5.76 + 1.53 × 10^−4^ ν_CTC_	1.06 × 10^2^
Energy, *hν* (eV)	Karipcin et al. [46]	From Tauc plot: (αhv)^2^ (eV cm^−1^)^2^ Against Energy (eV)	2.7036
Resonance energy, R_N_ (eV)	Briegleb and Czekalla [51]:	R_N_ = 7.7 × 10^−4^/[(h ν_CTC_/R_N_)–3.5]	0.7705
Transition dipole moment, µ _(Debye)_	Tsubumora and Lang [53]:	µ = 0.0958 [ε_CTC_ × Δν12/Δν]^12^	0.14 × 10^2^
Oscillator strength, *f*	A.B.P. Lever [52]	*f* = 4.32 × 10^−4^ ∫ ε_CTC_^dν^12	0.4990
Standard free energy change, ΔG^0^ (J mol^−1^)	Benesi-Hildebrand [26]	ΔG^0^ = −2.303 RT log *K*	−9.811

**Table 3 medicina-59-00756-t003:** Atom numbers, types, and their calculated charges on the energy minimized LOR molecule.

Atom Number	Atom Type	Charge	Atom Number	Atom Type	Charge
C(1)	Aromatic C, in benzene	0.0825	C(20)	Aromatic C, in benzene	−0.15
C(2)	Aromatic C, in benzene	0.41	C(21)	Aromatic C, in benzene	−0.1435
N(3)	Aromatic N, with s lone pair	−0.62	C(22)	Alkyl C, SP3	0.4235
C(4)	Aromatic C, in benzene	0.16	O(23)	Alcohol, ether O	−0.3625
C(5)	Aromatic C, in benzene	0	C(24)	Alkyl C, SP3	0
C(6)	Aromatic C, in benzene	−0.15	N(25)	Enamine, aniline N	−0.9
C(7)	Aromatic C, in 5-ring	0	C(26)	Cyano C	0.5371
C(8)	Aromatic C, in 5-ring	−0.1316	N(27)	Triple bond N	−0.5571
N(9)	Aromatic N, in 5-ring	0.314	C(28)	Alkyl carbon, SP3	0.2556
N(10)	Aromatic N, in 5-ring	−0.7068	F(29)	Fluorine	−0.19
C(11)	Aromatic C, in 5-ring	0.1078	C(30)	Alkyl carbon, SP3	0.3001
C(12)	Alkyl C, SP3	0.4811	H(31)–H(32)	H attached to C	0.15
N(13	Amide N	−0.6602	H(33)–H(34)	H attached to C	0
C(14)	Amide carbonyl C	0.5438	H(35)–H(37)	H attached to C	0.15
O(15)	Carbonyl O, in amide	−0.57	H(38)–H(41)	H attached to C	0
C(16)	Aromatic C, in benzene	0.0862	H(42)–H(43)	H of enamine N	0.4
C(17)–C(18)	Aromatic C, in benzene	−0.15	H(44)–H(49)	H attached to C	0
C(19)	Aromatic C, in benzene	0.19			

**Table 4 medicina-59-00756-t004:** Calibration parameters for LOR quantitation by MW-SPA via formation of colored CTC with DDQ.

Parameter	Value
Linear dynamic range (µg/well)	5–200
Intercept (a)	0.008
Standard deviation of intercept (SD_a_)	0.0033
Slope (b)	0.006
Standard deviation of slope (SD_b_)	0.0004
Correlation coefficient (r)	0.9996
Limit of detection (LOD, µg/well)	1.8
Limit of quantitation (LOQ, µg/well)	5.5

**Table 5 medicina-59-00756-t005:** MW-SPA precision at various LOR concentration levels.

Taken Concentration (μg/well)	Precision: Relative Standard Deviation (%)		Accuracy: Recovery (% ± SD) ^a^
	Intra–Assay, *n* = 3	Inter–Assay, *n* = 6	
6.25	0.93	1.56	101.2 ± 0.84
12.5	0.15	1.25	99.6 ± 1.25
25	0.53	1.44	98.8 ±1.51
50	0.86	1.12	100.5 ± 0.89
100	0.56	1.80	103.1 ± 2.12
200	1.76	1.86	101.4 ± 1.58

^a^ Values are mean of three determinations.

**Table 6 medicina-59-00756-t006:** Analysis of lorbrena^®^ tablets by the proposed MW-SPA.

Nominal Concentration (μg/mL)	Found Concentration ^a^ (μg/mL)		Recovery ^a^ (%)
62.5	63.22		101.2 ± 2.1
500	507.18		101.4 ± 1.4
1000	1032.03		103.2 ± 1.1
2000	1993.56		99.2 ± 1.5
		Mean	101.3
		SD	1.64

^a^ Values are mean of three determinations.

## Data Availability

All data are available from the corresponding author.

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
