# Peer review of "Novel High-Throughput Microwell Spectrophotometric Assay for One-Step Determination of Lorlatinib, a Novel Potent Drug for the Treatment of Anaplastic Lymphoma Kinase (ALK)-Positive Non-Small Cell Lung Cancer"

_medicina, 2023, doi:10.3390/medicina59040756_

Round 1

Reviewer 1 Report

I am writing about the paper entitled Novel High-Throughput Microwell Spectrophotometric Assay for One-Step Determination of Lorlatinib, A Novel Potent Drug for the Treatment of Anaplastic Lymphoma Kinase (ALK)-Positive Non-Small Cell Lung Cancer. The overall scientific idea behind this is very interesting how a spectrophotometric assay will help to understand the dose response during non-small cell lung cancer treatment. I would highly recommend this paper.    

Reviewer 2 Report

In this study, authors developed a high-throughput microwell spectrophotometric assay to determine patent lorlatinib. According to the results, this method looks reliable and can be potentially used in pharmaceutical industry. This manuscript was well-organized and written. I have no critical comment about it.  

Reviewer 3 Report

overall good rating

Figure 1: it is unclear 1) why the comparison of drug absorbance spectrum was specifically done with that dye with that dye, 2) mainly why they do not show the whole spectrum and 3) why there is initial noise

analytical publications describe methods and they do not necessarily need a mechanism

Figure 4: they characterize the complex and see a 1:1 ratio however their proposed mechanisms is just speculative, there are no  NMR , or IR spectra, they have not studied anything in that respect.

the dynamic range did not defined the order of magnitude, the minimum value used is not explained. maybe that's just enough to have the dosage in the pills, however it is very generic.

a true validation has not been performed: they did not verify their data with two independent methods, this must be clearly specified

the only method that exists for dosing is HPLC mass, as they stated but they need to update the reference also including the novel 2023 paper that describes dosing in cerebrospinal fluids

they must explained well and clearly that their purpose with these detection limits is only for quantification in tablets but not in body fluids. this has to be stressed
